# Utilization of Water-Soluble Aminoethylamino–β–Cyclodextrin in the Pfitzinger Reaction—Catalyzed to the Synthesis of Diversely Functionalized Quinaldine

**DOI:** 10.3390/polym12020393

**Published:** 2020-02-09

**Authors:** Yohan Kim, Vijay Vilas Shinde, Daham Jeong, Seunho Jung

**Affiliations:** 1Department of Systems Biotechnology & Department of Bioscience and Biotechnology, Microbial Carbohydrate Resource Bank (MCRB), Center for Biotechnology Research in UBITA (CBRU), Konkuk University, Seoul 05029, Korea; 2Institute for Ubiquitous Information Technology and Applications (UBITA), Center for Biotechnology Research in UBITA (CBRU), Konkuk University, Seoul 05029, Korea; vijay.shinde040@gmail.com

**Keywords:** aminoethylamino-β-cyclodextrin, supramolecular catalyst, Pfitzinger reaction, host–guest complexation study, one-pot synthesis

## Abstract

In this study we describe the use of an aminoethylamino-β-cyclodextrin (AEA–β–CD) as a supramolecular homogeneous catalyst for the synthesis of a series of diversely substituted quinaldine derivatives which are medicinally important, via Pfitzinger reaction. This supramolecular catalyst exhibited remarkable catalytic activity with high substrate scope to achieve the synthetic targets in good to excellent yield, 69–92%. The structural and morphological properties of the synthesized AEA–β–CD were determined through MALDI–TOF mass spectrometry, NMR, FT-IR, and SEM analysis. Possible reaction mechanisms were determined through molecular host–guest complexation and proposed based on 2D NMR (ROESY) spectroscopy, FT-IR, FE-SEM, and DSC.

## 1. Introduction

Quinaldine is an important structural unit which exists in natural products and biologically active molecules [1]. Their biological activities include antitubercular, antiprotozoal, antipsychotics, anticancer, antioxidant, anti-inflammatory, anti-HIV, and antifungal ones [2,3,4,5,6,7,8]. In addition, quinaldine and its derivatives are found in several commercially available drugs. For instance, 2-(2-fluorophenyl)-6,7-methylenedioxy quinolin-4-one monosodium phosphate (CHM-1-P-Na) is a preclinical anticancer agent, which showed excellent antitumor activity in an SKOV-3 xenograft in a nude mouse model [9,10]. For the synthesis of quinaldine, numerous name reactions have been reported, such as Skraup synthesis; Knorr quinoline synthesis; the Pfitzinger reaction; Combes quinoline synthesis; the Povarov reaction; the Doebner reaction; the Doebner–Miller reaction; the Gould–Jacobs reaction; Conrad–Limpach synthesis; Reihm synthesis; Friedlander synthesis; Niementowski quinoline synthesis; Camps quinoline synthesis; and Meth–Cohn synthesis [11]. Unfortunately, several of these reactions suffer from one or more drawbacks, such as harsh reaction conditions, hazardous organic solvents, low product yields, tedious workup procedures, relatively long reaction times, and difficulties in recovery and reusability of the catalysts. Consequently, the exploration continues to find better catalysts for the synthesis of functionalized quinaldine ring fragments in terms of operational simplicity, reusability, economic viability, selectivity and biodegradability.

Supramolecular chemistry denotes the chemistry beyond the molecule involving noncovalent intermolecular forces and has a profound influence on the catalysis of a variety of organic reactions to attain higher selectivity mimicking biological phenomena. A major application of supramolecular chemistry is the design and understanding of catalysts and catalysis. Noncovalent interactions are extremely important in catalysis, binding reactants into conformations suitable for reaction and lowering the transition state energy of reaction. Among the various known supramolecular catalysts, β-cyclodextrin (β-CD) is most preferable due to its unique properties, and for economic reasons when compared with α- and γ-cyclodextrins. Due to the hydrophobic cavity of β-CD, it allows “host–guest” inclusion complex formation with a wide variety of guest molecules through hydrophobic interaction, and the hydrophilic surface of β-CD makes the complex water-soluble. Modified β-CD provides novel derivatives with unique structures and functions to support the development of different research fields [12]. Additionally, amino-functionalized β-CD has been used as a catalyst in organic synthesis due to its powerful catalytic activity such as in multicomponent reaction (MCR) condensation, oxidation, and asymmetric synthesis [13,14,15].

As part of our continuing interest in modified β-CD catalyzed organic synthesis [16,17], herein we report an efficient and eco-friendly procedure for the synthesis of quinaldine derivatives from isatin, 1,3-dicarbonyl, and alcohol in water via Pfitzinger reaction catalyzed by AEA–β–CD, a reusable homogeneous catalyst, at 60 °C (Scheme 1). To the best of our knowledge, the synthesis of diversely functionalized quinaldine, catalyzed by AEA–β–CD, has not been reported yet.

## 2. Materials and Methods 

### 2.1. Chemicals

All chemicals were obtained from commercial suppliers (Aldrich and Alfa Aesar chemical companies, St. Louis, MO, USA), and ultrapure water was used. Water was purified using a Direct-Q Millipore water purification system from SAM WOO S&T Co., Ltd, Seoul, Korea. TLC was carried out using silica gel 60 F254 (Merck) plates. NMR spectra were recorded in parts per million (ppm) in CDCl_3_ and D_2_O on a Bruker Avance spectrometer (Karlsruhe, Germany) at 500 MHz using TMS as an internal standard.

### 2.2. Synthesis of mono-6-deoxy-6-aminoethylamino-β-cyclodextrin (AEA–β–CD)

Mono-6-deoxy-6-aminoethylamino-β-cyclodextrin (AEA–β–CD) was prepared using a previously detailed method [18].

### 2.3. General Procedure for the Synthesis of Quinaldine

AEA–β–CD (1176.43 mg, 1 mol %) was added to the suspension of isatin (1, 147.13 mg, 1 mmol) in water (4 mL) and it was then stirred at room temperature (RT) for 10 min; 1,3-dicarbonyl (2, 158.19 mg, 1 mmol) and alcohol (3, 46.07 mg, 1mmol) were added to the above dark red solution and heated to 60 °C till the completion of reaction. Reaction was monitored with TLC. After completion, the reaction mixture was cooled to room temperature and extracted with ethyl acetate. Crude compound was purified by column chromatography (Hexane 80: Ethyl acetate 20). The AEA–β–CD catalyst was recovered from the aqueous fraction by precipitation with acetone (15 mL). The catalyst was filtered, washed with acetone (3 × 10 mL), dried under vacuum at 70 °C for 9 h, and reused.

### 2.4. Matrix Assisted Laser Desorption/Ionization-Time of Flight (MALDI–TOF) Mass Spectrometry

The sample solution contained 1 μL of AEA–β–CD and 1 μL of 2,5-dihydroxybenzoic acid (DHB) solution (10 mg/mL DHB/H_2_O solution) as a matrix. The dried sample was analyzed with the AB SCIEX MALDI TOF-TOF 5800 System (Applied Biosystems, Foster city, CA, USA). All mass spectra were obtained in positive linear mode with an accelerating voltage of 20 kV in the reflector mode.

### 2.5. Nuclear Magnetic Resonance (NMR) Spectroscopy 

^1^H, ^13^C NMR, and HSQC spectra were recorded with a Bruker Avance 500 (Karlsruhe, Germany) spectrometer. The samples solutions were made up in deuterated water (D_2_O, 99.96%), and NMR measurements were performed with 600 μL samples in 5 mm NMR tubes.

### 2.6. Fourier Transform-Infrared (FT-IR) Spectroscopy 

The Fourier-transform infrared (FT-IR) spectra were obtained using a Bruker IFS-66/S spectrometer (AMX, Germany) with potassium bromide (KBr) pellets as support in the scanning range of 650–4000 cm^−1^.

### 2.7. Rotating Frame Nuclear Overhauser Spectroscopy (ROESY)

The 2D ^1^H—^1^H ROESY spectrum of the inclusion complex of AEA–β–CD with 5-chloroisatin (CI) was recorded using 256/2048 complex data points and a pulse train to attain a spin-lock field with a mixing time of 400 ms for the complex. The NMR analyses were carried out on the Bruker Avance 600 MHz spectrometer in D_2_O solvent at 25 °C.

### 2.8. Field Emission Scanning Electron Microscopy (FE-SEM)

Hitachi S-4700, produced by the Hitachi High-Technologies Corporation, was used to carry out the field emission scanning electron microscopy (FE-SEM). To fix the samples on a brass stub, a double-sided adhesive carbon tape was used. The powder samples were coated by a thin gold layer on the surface. The images were photographed at an excitation voltage of 5 kV.

### 2.9. Differential Scanning Calorimetry (DSC)

The thermal behaviors of 5-chloroisatin, AEA–β–CD, their physical mixture, and the inclusion complex were then examined using a DSC 7020 (SEICO INST., Chiba, Japan). A sample of 5 mg was placed into a sealed aluminum pan prior to heating under nitrogen (40 mL min^−1^) at a scanning rate of 10 °C min^−1^. The observations were recorded over the temperature range of 50 to 400 °C. 

### 2.10. Phase Solubility Analysis

Due to the low water solubility of isatin, it was dissolved in a solution (1 mL) of acetone: triply distilled water (TDW) in a ratio 4:1. To adjust different concentrations of AEA–β–CD (0, 4, 8, 12, 16, and 20 mM), AEA–β–CD was added to the isatin solution (20 mM). The suspensions were magnetically stirred at 25 °C for 24 h, and protected from light to prevent the decomposition of the molecules. After equilibrium was reached, acetone was evaporated; then, the mixture was lyophilized. The lyophilized sample was dissolved in water and filtered using a polyvinylidene fluoride (PVDF) 0.2-mm filter (Whatman). Each sample was analyzed using a spectrophotometer (UV2450, Shimadzu Corporation, Kyoto, Japan) at a wavelength of 295 nm to measure the dissolved isatin concentration. The graphs of concentrations of isatin and AEA–β–CD were plotted using the obtained data. The binding constant, $K_{c}$, for the complex formation was calculated from the linear portion of the solubility diagram using the Higuchi and Connors equation (Equation (1)) [19].
(1)$$K_{c}=\frac{\mathrm{Slope}}{S_{0}\left(1-\mathrm{Slope}\right)}$$

## 3. Results and Discussion

### 3.1. Characterization of AEA–β–CD

The catalyst mono-6-deoxy-6-aminoethylamino-β-cyclodextrin (AEA–β–CD) was prepared in two concise steps which are depicted in Scheme 2.

Synthesized AEA–β–CD was well characterized by NMR spectroscopy (Appendix A) and MALDI TOF mass spectrometry. The pseudo-molecular ion peak at m/z = 1177 as [AEA–β–CD+H]^+^ is shown in its MALDI–TOF mass spectrum (Figure 1).

The heteronuclear single quantum coherence (HSQC) spectrum was also used to confirm the identity of the synthesized AEA–β–CD. In Appendix A, the x-axis shows the ^1^H NMR spectrum, where the H1 to H6 protons of the glucopyranose units are assigned at 5.05, 3.64, 3.84, 3.57, 3.93, and 3.86 ppm. The substituted ethylene protons of H7 and H8 appeared at 2.76 and 2.73 ppm. Furthermore, the 2:7 integral ratios of the H8 protons to the H1 protons confirmed the mono-functionalized β-CD. The y-axis has the ^13^C NMR chemical shifts of the AEA–β–CD, which are assigned as C1-C6, 101.84, 72.09, 73.10, 81.15, 72.09, and 60.31. The substituted C7 and C8 carbon appeared at 39.36 and 49.01 ppm, respectively.

The IR spectrum of AEA–β–CD contains the characteristic absorption band for the spectrum with O–H stretching at 3390 cm^−1^, sp^3^ C–H stretching around 2929 cm^−1^, and C–O stretching around 1028 cm^−1^. The N–H bending vibrations were observed at 1639 and 1576 cm^−1^ (Figure 2).

### 3.2. Optimization of Reaction Conditions with AEA–β–CD as a Catalyst

In order to standardize the catalytic effect of AEA–β–CD for the synthesis of quinaldine derivatives via Pfitzinger reaction, we carried out the reaction between isatin (**1a**), ethyl 3-oxohexanoate (**2a**), and ethanol (**3a**) as a model reaction in water without catalyst (Scheme 3). The reaction did not take place in the absence of the catalyst (Table 1, entry 1). These results showed the requirement of the catalyst. Next, we investigated the reaction using AEA–β–CD (1 mol %) under aqueous conditions at 50 °C. Under these conditions, the desired product **4a** was obtained in 51% yield (Table 1, entry 4). Among the various catalysts tested for this reaction, AEA–β–CD was found to be the most efficient catalyst, which gave diethyl 2-propylquinoline-3,4-dicarboxylate 4a in 51% yield with water as a suitable solvent. To obtain a higher yield, the loading of catalyst was checked, and we observed that low loading of catalyst AEA–β–CD could not complete conversion of reactant to product (Table 1, entry 10). The effect of solvent (protic and aprotic) on the model reaction was studied by using 1 mol % of AEA–β–CD in different solvents (Table 1, entry 12–15). The reaction was also studied at various temperatures (40, 60, and 80 °C). 1 mol % of AEA–β–CD gave the highest yield of diethyl 2-propylquinoline-3,4-dicarboxylate 87 % (Table 1, entry 17). Phase solubility diagrams of isatin with AEA–β–CD were obtained using the UV spectra of different sample concentrations (Appendix A). The plot of isatin and AEA–β–CD appeared to be A_L_ type, showing direct proportionality between the concentration of AEA–β–CD and solubility of isatin. The linear graph derived from isatin and AEA–β–CD suggested a 1:1 molecular association. The corresponding binding constants, K_c_, were determined using Equation (1) depending on the reaction temperatures (Table 2). At 60 °C, the binding constant value was sufficiently increased compared to other temperatures. Besides, we calculated the turnover number for aminoethylamino-β-cyclodextrin catalyst, and the TON of entry 17 with the highest yield was 87 (TON = number of moles of reactant consumed /moles of catalyst). Recovered yield upon each cycle was shown in Appendix A. We did not observe any significant mass loss of catalyst during the reaction cycles. Since β-CD is a supramolecular catalyst; it solubilizes reactants in water through host–guest complex formation [20]. This homogenization of reactants in the solvent increases their interaction with the catalyst, and therefore improves the rate of reaction and yield of the final product. Although the reactants have good solubility in ethanol, water acts as a good reaction medium due to the easy separation of the product and higher rate of product formation **4a**. Thus, AEA–β–CD speeds up the rate of the reaction. After extraction of the product, the aqueous media containing the dissolved AEA–β–CD could be reused for the next set of reactions. Thus, the optimum conversion of reactants to product was achieved with 1 mol % of AEA–β–CD in aqueous conditions at 60 °C. In addition, a strong cavity binder, adamantane carboxylic acid, was added to AEA–β–CD first to make the inclusion complex sufficiently, and then, the reaction was performed. We observed no catalytic activity in this reaction (Table 1, entry 19). This result proves synthesis of quinaldine derivatives via Pfitzinger reaction occurs in the cavity of the CD.

### 3.3. Optimization of Reaction Conditions for Synthesis of Quinaldine Derivatives

With the optimal reaction conditions in hand, the scope of substrates was then explored. The results are summarized in Table 3. To prepare methylquinoline-4-carboxylate derivatives with a variety of different substituents on the benzene ring of the quinaldine nucleus, several isatin bearing an electron-donating or withdrawing group on the benzene ring were reacted with several 1,3-dicarbonyls and different alcohols under optimized reaction conditions (1 mol % of AEA–β–CD, water, 60 °C). To our delight, we found this conversion to be very general for a wide range of isatin, 1,3-dicarbonyls, and different alcohols; it provided easy access to densely substituted quinaldine **4**. Halogenated (5-Br, 5-Cl, 5-F) isatin and electron-neutral (5-H) isatin were smoothly converted to the corresponding products **4a**-**4w** in good to excellent yields (77–92%). When the substituent of isatin was changed to strong electron-withdrawing groups (5-NO_2_ and 7-F), moderate yields were obtained (Table 3, entry 5 and 18). On the other hand, isatin being substituted with electron-donating groups (-OMe) was tolerated well and gave excellent yields (Table 3, entry 3, 8, 17, 23). As shown in Table 3, all 1,3-dicarbonyls also proceeded smoothly and afforded the desired products in excellent yields. 

All the compounds were characterized by ^1^H NMR, ^13^C NMR, and HRMS (Appendix A).

Furthermore, the reutilization performance of the AEA–β–CD catalyst was confirmed, as AEA–β–CD was recovered simply as precipitated from the aqueous layer, by the addition of acetone. The recovered catalyst was filtered, washed with acetone, and dried under vacuum at 70 °C for 9 h, after which it could be reused. The reusability of AEA–β–CD was verified over four reaction cycles. The recyclability of the catalyst was monitored, with reaction yields of 87%, 87%, 85%, and 85%, in the successive cycles (Table 1, Entry 17), which proved the reusability of the catalyst.

### 3.4. ROESY Spectroscopy of AEA-β-CD/CI Inclusion Complexes

To further examine the role of the AEA–β–CD catalyst in the reaction, the inclusion complexation of the reactants and AEA–β–CD was investigated. 2D NMR ROESY was used as the ultimate characterization tool for the elucidation of complexation between isatin and AEA–β–CD [21]. As shown in Figure 3, the protons on the 5-chloroisatin (CI) showed clear NOE interactions with the H3 and H5 protons of AEA–β–CD. Moreover, the ROESY experiment explained the inclusion mode of the AEA–β–CD/CI complex. The spectrum exhibited typical NOE interaction between the aromatic protons of CI at 7.60 and 7.00 ppm, and the inside cavity’s H3 and H5 protons of AEA–β–CD were detected at 3.84 and 3.93 ppm, respectively (see the enlarged view in Figure 3). Additionally, NOE linking between the H1 proton of AEA–β–CD and CI was not observed, as it is well-known that the H1 proton exists on the external part of the AEA–β–CD cavity. Thus, the role of β-CD-fragment of AEA–β–CD was confirmed to be solubilizing reactants in water through host-guest complex formation and improving the rate of reaction.

### 3.5. Fourier Transform Infrared (FT-IR) Spectroscopic Analysis

Additional supportive evidence of complex formation was established using FT-IR spectroscopy [22]. In Figure 4, the FT-IR spectrum of 5-chloroisatin contained prominent absorption bands of the NH group (3241 cm^−1^); strong bands at 1747 cm^−1^, attributable to the carbonyl groups; and a third strong band at 1615 cm^−1^, attributable to aromatic C=C stretching vibrations. AEA–β–CD shows characteristic absorption band for O–H stretching at 3386 cm^−1^, sp^3^ C–H stretching around 2925 cm^−1^, and C–O stretching around 1028 cm^−1^. The N–H bending vibrations were observed at 1642 cm^−1^ and 1572 cm^−1^. In the physical mixture, characteristic peaks of 5-chloroisatin were still detected, but with low intensity, which indicates that a weak interaction occurs between 5-chloroisatin and AEA–β–CD for forming the inclusion complex. The disappearance of the characteristic 5-chloroisatin peaks at 3241, 1747, and 1615 cm^−1^ in the lyophilized complex can be attributed to inclusion of these functional groups inside the AEA–β–CD cavity.

### 3.6. Field Emission Scanning Electron Microscopy (FE-SEM) Analysis

The FE-SEM characterizes the morphological changes of the inclusion complexes [23]. The SEM images of AEA–β–CD, 5-chloroisatin, the 5-chloroisatin/AEA–β–CD physical mixture (PM), and the 5-chloroisatin/AEA–β–CD complex are shown in Figure 5. AEA–β–CD (a) appeared as an amorphous plate while 5-chloroisatin (b) was an irregular crystalline particle. The 1:1 physical mixture (c) involved a bulky particle (AEA–β–CD) with characteristic 5-chloroisatin crystals observed beside its surface. Hence, no interaction took place in the physical mixture in the solid state. However, a drastic change in the structural morphology of the lyophilized complex (c) was observed. This indicates the existence of a new solid phase, probably due to the development of inclusion complexes. This also suggests the homogenous distribution of two components.

### 3.7. Differential Scanning Calorimetry (DSC) Analysis

DSC is a valuable technique with which to characterize the solid-state interactions of a host and guest [15]. The thermal behaviors of AEA–β–CD, 5-chloroisatin, the 5-chloroisatin/AEA–β–CD physical mixture, and the 5-chloroisatin /AEA–β–CD complex were investigated using DSC, as shown in Figure 6. A sharp endothermic peak was observed in the DSC curve of 5-chloroisatin, at 252 °C, corresponding to its fusion peak. The DSC curve of AEA–β–CD contained an endothermic peak at approximately 231 °C. For the physical binary mixture of 5-chloroisatin/AEA–β–CD, the endothermic peak of 5-chloroisatin was observed at 250.5 °C, with a small shift, which can be attributed to the solid–solid interaction. However, in the DSC curve of the 5-chloroisatin /AEA–β–CD complex, the endothermic peak corresponding to free 5-chloroisatin disappeared. This indicates that AEA–β–CD successfully formed complexes with isatin, which changed its physical properties.

### 3.8. Synthetic Mechanism of Diversely-Substituted Quinaldine Derivatives with AEA–β–CD as a Catalyst

The mechanism for the AEA–β–CD catalyzed formation of diversely-substituted quinaldine explained in Scheme 4. The hydrogen bond was made between the oxygen of isatin and the amine group of CD. According to electronegativity, this imbalance of electrons is represented by delta negative (δ-) on the oxygen and delta positive signs (δ+) on the amine group. This increases the electron-withdrawing properties of the oxygen. Therefore, the reaction could occur easier by OH attack. Isatin **1** is catalyzed by AEA–β–CD and converted to isatinic acid **A**, which condenses with 1,3-dicarbonyl **3** with the release of water, and that is followed by an intramolecular cyclization to give quinolone-4-carboxylic acid intermediate **B**. There is a further nucleophilic attack by ethanol on the acid group of intermediate **B** to get the final compound **4.** The binding constant (Kc) for the complex between isatine and AEA–β–CD was calculated from the linear portion of the solubility diagram using the Higuchi and Connors equation [19] in Appendix A.

## 4. Conclusions

In conclusion, an efficient and environmentally benign method has been developed using AEA–β–CD as a homogeneous supramolecular catalyst for the synthesis of diversely substituted quinaldine derivatives via Pfitzinger reaction. AEA–β–CD is a supramolecular catalyst, and thus it enhances the solubility of the reactants in water by trapping the substrate molecules in the cavity of cyclodextrin. This methodology has the returns of mild reaction conditions, an inexpensive biodegradable catalyst, and a green solvent.

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
