# Peer review of "Utilization of Water-Soluble Aminoethylamino–β–Cyclodextrin in the Pfitzinger Reaction—Catalyzed to the Synthesis of Diversely Functionalized Quinaldine"

_polymers, 2020, doi:10.3390/polym12020393_

Round 1

Reviewer 1 Report

This manuscript by Jung and coworkers reports on the synthesis for monofunctioalized beta-cyclodextrin (beta-CD), and its application as a supramolecular catalyst for Pfitzinger reaction. The amine-based derivative of beta-CD showed an excellent catalytic activity for this type of reaction, and performed well with a wide spectrum of substrates. Overall, the topic is very interesting, and the obtained results are supported by several spectroscopic techniques in a well-presented format. However, minor revisions are required before the acceptance of the manuscript.

Minor revisions:

It is very important to show that the catalyst can be further used for several reaction cycles, however, I wonder if the whole amount of the catalyst is being recovered upon each cycle or there is a mass lose. One of the major challenges in supramolecular catalysis is the product inhibition, in which the product blocks the cavity of the macrocycle, and therefore prohibit further catalytic turnover. In this system, however, it is not the case for this system. The authors showed further elaborate on the reaction mechanism and discuss the possible reason for not having a product inhibition. As the reaction is done in water, the author showed pay more attention to the pH, for which the amino-group could be either protonated (NH3+) or neutral (NH2), and it might turn out that both have different catalytic behavior. In addition, the authors should comments on the importance of having an amine group in enhancing the reactivity. An additional control experiment could be performed to proof that the reaction indeed occurs inside that CD cavity, that is, the reaction could be done with a CD-blocked cavity by adding a strong cavity binder, such as adamantane carboxylic acid, which know to have a high binding affinity to beta-CD; no catalytic activity showed be observed. The synthesis of mono-6-deoxy-6-aminoethylamino-β-cyclodextrin has already been published by the same authors in 2017 (Choi, J. M.; Park, K.; Lee, B.; Jeong, D.; Dindulkar, S. D.; Choi, Y.; Cho, E.; Park, S.; Yu, J.-h.; Jung, S. Carbohydr. Polym. 2017, 163, 118). In case the same procedure was used in this manuscript. I suggest to just referring to original article to avoid duplication, and remove the synthesis part.

Other comments are given below:

Line 73: change “melt” to “dissolved”

Line 87: the sentence is incomplete

Section 2.4: the mass units of the reactants are missing

Scheme 2: change the weird symbol after 60 to °C

Scheme 3: please rescale the chemical structures (enlarge them) to match the CD representation.

Reviewer 2 Report

The work presented in this article describes the synthesis of quinaldine derivatives following Pfitzinger reaction in the presence of β-cyclodextrin (β-CD) equipped with tethered ethylene diamine moiety. β-CD is acting as a host for isatin substrate and it is suggested to catalyze the hydrolysis of isatin’s amide bond to give a keto-acid intermediate that is readily available for condensation with dicarbonyls under aqueous conditions.

I am a bit puzzled by the authors' intention to publish this work in the field of polymers. This work has nothing to do with polymers, neither polymer synthesis nor polymer properties and applications. This work is more suitable for publication in journals specialized in supramolecular chemistry or organic synthesis.

I also recommend to be more modest with the title: “Utilization of biodegradable aminoethylamino-β-cyclodextrin…” The biodegradability of the catalyst was not studied.

Other minor issues:

In Table 1, the performance of aminoethylamino-β-cyclodextrin is compared to that of inorganic bases and piperidine (entries 6-9). This comparison is based on that the bases are treated as catalyst. However these bases are supposed to operate at stoichiometric range and should not be considered as catalysts. The authors should provide the TON for aminoethylamino-β-cyclodextrin catalyst. The use of alcohol is a post-modification reaction to the Pfitzinger reaction. It is not affecting the main catalytic reaction. In the suggested mechanism, the authors claim that the amino group in the aminoethylamino-β-cyclodextrin catalyst coordinates to the isatin and accelerates the hydrolysis of the amide bond. So a non-covalent interaction is involved. However, under the reaction conditions (60 C to 80 C), this interaction cannot hold. The authors should give a proof for such a strong interaction. To my opinion this is a simple base-catalyzed reaction that form a β-keto amide, which then is attacked by intramolecular imine to form six-membered ring.

All together, the article should be rejected on grounds of suitability, but if authors consider another suitable journal then it may be accepted after addressing to the above comments.

Reviewer 3 Report

The manuscript describes the preparation of diversely substituted quinaldine derivatives using aminoethylamino-functionalized β-cyclodextrin as a green, reusable and effective supramolecular catalyst. In addition, the authors have characterized the catalyst, the prepared quinaldine products and the inclusion complex that the catalyst is able to form with 5-chloroisatin (one of the isatin derivatives that were used).

The study is of interest, solid, well presented and therefore it could be published in Polymers after minor revision. Here are some points that should be amended.

Scheme 1, “R4OH” should be “R4OH”. Line 70, “CDCl3” and “D2O” should be “CDCl3” and “D2O” Line 73, it would be better “was dissolved” instead of “was melt” Line 80, it should be “high vacuum-drying” Line 84-87, the text should be amended, some parts are missing Line 119, were the samples coated with a thin gold layer? The sentence is not clear Line 152, please replace “C=O” with “C-O” Line 153, N-H stretching vibrations appear at much higher wavenumbers. Maybe, those are N-H bending vibrations. Figure 2, “Cm-1” should be “cm-1” Line 167, “catalyst” should be plural Line 171, “was studied in using” should be amended Before table 2, scheme 1 appears a second time. One of the two copies could be removed Line 251, it could be N-H bending, not stretching vibrations Were the physical mixture and the inclusion complex prepared using the same catalyst:5-chloroisatin 1:1 molar ratio? Figures 4 and 6, if the overlapped curves are split, then the values on the y axis should be deleted Figure 6, DSC analysis should be performed below the decomposition temperature of samples and the thermal degradation of cyclodextrins usually occurs in the 300-350 °C range. TGA analysis would help to better understand the DSC curves The authors should explain why β-cyclodextrin was preferred over α- and γ-cyclodextrin

Round 2

Reviewer 2 Report

This new version by Seunho Jung and co-workers is better, yet some scientific issues in the manuscript that were raised previously are still not convincing. I’ll therefore repeat them, perhaps being more specific will help. After addressing to these issues in the revised MS it can be published.

I know that β-CD is biodegradable and the argument is on that this property has nothing to do with this research. You cannot specify in the title a property just to give it attractiveness without showing its relevance to the study. I prefer that the title would be “Utilization of water-soluble aminoethylamino-β-cyclodextrin in Pfitzinger reaction:…” Thus, directing the readers to the fact that this β-CD derivative operates in Pfitzinger reaction as a catalyst under aqueous conditions. The TON values in Table 1 are very wrong! Turnover Number (TON) = (number of moles of reactant consumed)/(mole of catalyst). For a catalyst at 1 mol-%, the max. TON = 1000 (if conversion of substrate is 100%), etc. Having TON ≤1 has no meaning to catalytic activity. For that, the authors should also provide the conversion-% of substrate. Moreover, TON should be provided in Table 1 just for the best result (optimized conditions: entries 17-18). Study that is more informative would be to see catalytic activity in the synthesis of quinaldine derivatives (Table 2). Please provide representative TON values in Table 2. The authors mentioned that at lower loading of catalyst (Table 1 entry 10) the reaction did not complete. The question is: why 1 mol-% works to give complete conversion and 0.5 mol-% does not. It seems to me that this is an issue of the binding constant between isatine and β-CD. If it is in the range of logK≈3 then it make sense but this data has to be presented. On the other hand, higher K values would be needed to allow inclusion at high temperature (60-80 C). Please provide the binding constant between b-CD and isatine under the same conditions since it also related to your proposed mechanism (first step: inclusion of isatine to b-CD cavity).

Author Response

Please see the attached the file.

Round 3

Reviewer 2 Report

I sincerely appreciate the authors' respond to my comments and they have made the necessary changes.

However, the TON values are related to yields and conversion. Only conversion is related to starting material consumption.

Assuming that ALL molecules of starting material were consumed (~1 mmol), then TON = 1 mmol/0.01 mmol catalyst = 100. If you state that in all reactions mentioned in Tables 1 and 2 the starting material was fully consumed, then TON is 100 for all reactions. In this case it is better to mention this value in the text and not in Tables.

The binding constant study is satisfying but you should provide more details on the technique, procedure description (NMR data of free isatin and bound) and what are the values in the graph: are they "soluble" isatin? how do you extract these values?
